behaviour

phonotaxis, *Gryllus bimaculatus*, calling song, oddball paradigm, tolerant pattern recognition, acoustic communication

**Author for correspondence:**
Berthold Hedwig
e-mail: bh202@cam.ac.uk

# Tolerant pattern recognition: evidence from phonotactic responses in the cricket *Gryllus bimaculatus* (de Geer)

Adam M. Bent and Berthold Hedwig

Department of Zoology, University of Cambridge, Downing Street, Cambridge CB2 3EJ, UK

AMB, 0000-0002-2679-2208; BH, 0000-0002-1132-0056

When the amplitude modulation of species-specific acoustic signals is distorted in the transmission channel, signals become difficult to recognize by the receiver. Tolerant auditory pattern recognition systems, which after having perceived the correct species-specific signal transiently broaden their acceptance of signals, would be advantageous for animals as an adaptation to the constraints of the environment. Using a well-studied cricket species, *Gryllus bimaculatus*, we analysed tolerance in auditory steering responses to '*Odd*' chirps, mimicking a signal distorted by the transmission channel, and control '*Silent*' chirps by employing a fine-scale open-loop trackball system. *Odd* chirps on their own did not elicit a phonotactic response. However, when inserted into a calling song pattern with attractive *Normal* chirps, the females' phonotactic response toward these patterns was significantly larger than to patterns with *Silent* chirps. Moreover, females actively steered toward *Odd* chirps when these were presented within a sequence of attractive chirps. Our results suggest that crickets employ a tolerant pattern recognition system that, once activated, transiently allows responses to distorted sound patterns, as long as sufficient natural chirps are present. As pattern recognition modulates how crickets process non-attractive acoustic signals, the finding is also relevant for the interpretation of two-choice behavioural experiments.

## 1. Introduction

Orthopterans are well-studied insects due to their use of conspicuous acoustic communication [1–4]. Through the stridulation of their wings, male field crickets produce a species-specific calling song, which females detect and if they are ready to mate, move toward males in a behaviour known as phonotaxis. In the bi-spotted field cricket, *Gryllus bimaculatus*, female phonotactic behaviour is sharply tuned to the calling song chirp and pulse pattern of conspecific males [5–9]. However, the calling song of male crickets is a long-distance signal which may become distorted in the transmission channel before being received [10–12]. Due to reverberations, the calling song's pulse pattern may become degraded within a short distance from the signaller, and conspecifics no longer respond to such distorted signals [13], as they become difficult to process by a pattern recognition system. A tolerant sensory system, which is adapted to the impact of the transmission channel and would accept distorted signals while processing the species-specific sound pattern, could be beneficial for females orienting in a complex environment. Here, we provide evidence for the presence of such a sensory system in female *G. bimaculatus*.

We investigated the propensity of crickets to respond to distorted non-attractive acoustic stimuli. To do so, experiments were designed analogous to 'oddball' paradigms used in psychological studies, where natural stimuli are presented with 'odd' portions inserted [14]. We exposed female crickets to different ratios of *Normal* and *Odd* chirps to infer the presence and timescale of tolerant sensory processes and examined the effect this may have on their

phonotactic behaviour. We tethered crickets so that they walked on an air-suspended trackball while sound patterns were presented from a speaker at the left or right side [14]. The rotations of the trackball were measured and revealed the female phonotactic steering behaviour to sound sequences with attractive and distorted non-attractive chirps combined at different ratios.

## 2. Material and methods

### (a) Animals
Last instar female *G. bimaculatus* were isolated from a colony at the Department of Zoology, Cambridge. Eclosed females were individually housed in plastic containers, they had access to food and water ad libitum, and were kept at a temperature of 25–28°C. Only unmated individuals older than 7 days post-eclosion were used as phonotactic responsiveness increases over the first week of adulthood [15]. One day prior to trials, the front wings of the test animals were cut, and an insect pin (approx. 32 mg) was waxed vertically onto the first abdominal tergite, close to the animal's centre of gravity. Females with the pin could move around freely and were returned to their containers prior to testing. Due to laboratory restrictions imposed by the COVID-19 pandemic, we used different cohorts of crickets over the course of the experiments.

### (b) Trackball set-up and recordings
Female phonotaxis to acoustic stimuli was measured through an open-loop trackball set-up. Females tethered to an insect pin were placed stationary on top of the lightweight air-suspended track ball, which they moved with their legs when walking. The forward–backward and left–right movements of the track-ball were captured with an optoelectronic sensor and indicated the walking speed and direction of the specimen. The lateral steering component was analysed as a reliable indicator of pho-notactic behaviour (see details in [14]). Females that showed a steering response below 50 mm to a *Normal* chirp sequence over the course of a minute were not included in analyses.

### (c) Auditory stimuli
Auditory stimuli were created with the software Cool Edit 2000 (Syntrillium, Phoenix, USA), with a sampling rate of 44.1 kHz, a 16-bit amplitude resolution and the sound intensity calibrated to 75 dB SPL. We created *Normal* calling song chirps similar to the natural song of *G. bimaculatus* [13], with a carrier frequency of 4.8 kHz. Each chirp consisted of five pulses with 20 ms pulse duration, including a 2 ms rise and fall, and a 40 ms pulse period, followed by a 175 ms chirp interval, resulting in a 375 ms chirp period. All acoustic stimuli were delivered via two speakers (Neo 13S, Sinus Live, Conrad Electronics, Hirschau, Germany) positioned 57 cm in front of the cricket at the left and right side at 45° to its long axis.

### (d) Experiment 1: identifying a distorted non-attractive *Odd* chirp
Crickets do not respond to conspecific calling songs with the pulse structure of chirps distorted by reverberations [13]. In order to create a distorted non-attractive chirp pattern, we designed artificial chirp stimuli with different amplitude envel-opes that maintained the same chirp period, duration and frequency as the *Normal* chirps, but had no pulse structure. The envelope shapes covered one chirp-long sound and were 'Rec-tangle', 'Oval', 'Crescendo' and 'Decrescendo' (figure 1a), they

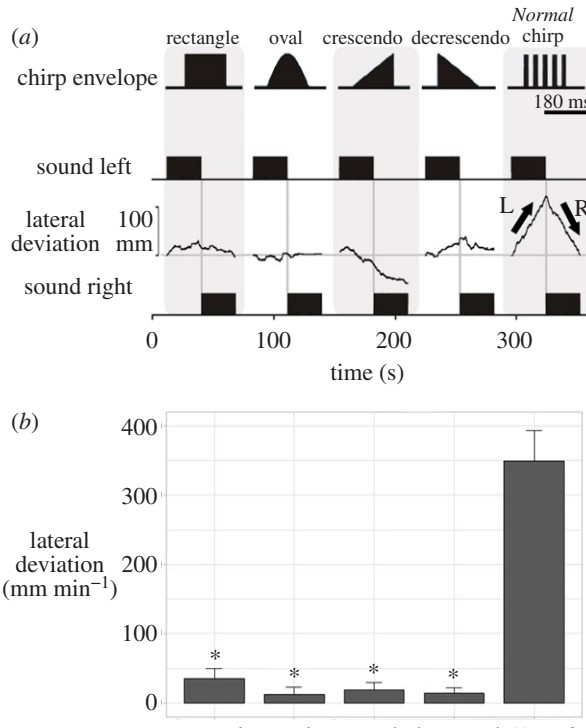

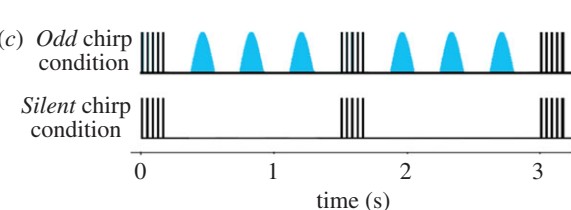

**Figure 1.** (*a*, top) Chirps with rectangle, oval, crescendo, decrescendo wave-form envelopes and a *Normal* chirp. (*a*, bottom) Phonotactic steering of a cricket to sound patterns composed of the different envelopes, 30 s sequences are presented alternating from the left and right. Arrows indicate steering to the left (L) and right (R), respectively. (*b*) Mean steering response to chirps with different envelopes or *Normal* chirps over the course of 1 min ($\bar{x} \pm$ s.e., $N = 7$). Responses with an asterisk are significantly different to the *Normal* chirp response in *post hoc* pairwise analyses. (*c*) Depiction of the *Odd* chirp and *Silent* chirp condition used in the 1 : 3 ratio; a *Normal* chirp is followed by three *Odd* chirps (blue) or *Silent* chirps. Sequences with *Normal*, and *Odd* or *Silent* chirps have the same temporal organization, but *Silent* chirps do not provide any sound. (Online version in colour.)

did not contain the pulsed pattern of normal chirps. Chirps with these envelopes were presented sequentially, for 30 s from the left then 30 s from the right, with 15 s silence between each, followed by a *Normal* chirp song pattern (figure 1a). A total of seven females were tested to select the *Odd* chirp with the lowest phonotactic response. The oval envelope chirp was the least attrac-tive (figure 1b) and was used as the *Odd* chirp in experiments going forward.

### (e) Experiment 2: phonotaxis to combinations of *Normal* chirps and *Odd* chirps
To test for tolerant phonotaxis behaviour, we combined *Normal* and *Odd* chirps at different ratios, i.e. a *Normal* chirp was fol-lowed by a certain number of *Odd* chirps (figure 1c). The ratios created were 1 : 19 (1 *Normal* chirp followed by 19 *Odd* chirps), 1 : 14, 1 : 9, 1 : 7, 1 : 5, 1 : 3 and 1 : 1. The phonotactic steering responses of females ($N = 27$) were tested. We subsequently

tested a different cohort of crickets ($N = 27$) to analyse the propensity of crickets to respond to *Odd* chirps when *Normal* chirps are more frequent. The ratios presented here were $1:3$ (one *Normal* chirp followed by three *Odd* chirps), $1:2$, $1:1$, $2:1$ and $3:1$. In both sets of experiments, an All *Odd* and All *Normal* chirp sequence were presented at the beginning or end of the tests.

As a control, we created test patterns with the same temporal organization, but we replaced the *Odd* chirps with a period of silence for the same duration (figure 1*c*), hereafter referred to as *Silent* chirps. This resulted in a corresponding number of stimulus sequences with increasing numbers of *Normal* chirps, with the addition of an All *Silent* chirp and an All *Normal* chirp sequence. Sound patterns were always presented from least to most occurrences of *Normal* chirps. Females were tested once with the *Odd* chirp and *Silent* chirp paradigms, with at least 24 h between each test. The paradigm which was tested first was selected pseudo-randomly. In all experiments the overall lateral deviation of the females toward the different acoustic test patterns was analysed.

## (f) Experiment 3: phonotactic steering velocity in response to *Normal* chirps and *Odd* chirps

To understand the dynamics of the behaviour, we analysed the lateral steering velocity of the phonotactic responses. Triggered by the start of the *Normal* chirps, we averaged the lateral steering velocity in response to the *Normal*, *Odd* and *Silent* chirps in certain ratios. The steering velocity provides the dynamics of the female behaviour at high temporal resolution and allowed us to compare the amplitude of the steering responses. Subsequently, we tested a final group of females ($N = 30$) to analyse if processing of *Normal* chirps would affect the response to *Odd* chirps presented from the opposite side. We used a ratio of *Normal* to *Odd* chirps at $1:1$. For comparison, we also tested females on a sequence with *Normal* chirps presented from alternating sides, and an All *Odd* sequence not presented from alternating sides.

## (g) Data analysis

Trackball movements were recorded using software designed in LabView v. 5.01 (National Instruments, London, UK). Off-line data analysis was done with Neurolab [16]. Each chirp type was presented for 30 s from the left and right. The steering velocity of the crickets was calculated based on the output of the optical sensor and the lateral deviation over each presentation was obtained by integrating the velocity data for the left and right sound presentation. These were pooled to obtain a measure for the phonotactic response over a 1 min window [17].

All statistical analyses were conducted using R Studio [18,19] and the packages 'dunn.test' [20] and 'PMCMR' [21]. Where necessary, data were tested for normality using Shapiro–Wilk tests, and non-parametric tests were used where appropriate. Graphs detailing results of the experiment were either exported from Neurolab, or created in R using the package ggplot2 [22].

The steering responses to chirps with different envelopes and *Normal* chirps (Experiment 1) were tested using a Friedman rank sum test paired with Conover *post hoc* tests with Bonferroni corrections. The same procedure was also used to test for differences in steering responses between chirp ratios in Experiment 2. Where appropriate, we used multiple Wilcoxon signed-rank tests and paired *t*-tests to test for within chirp ratio differences between *Odd* and *Silent* chirp conditions and between the same ratios presented in different paradigms. In Experiment 3, we used Kruskal–Wallis tests paired with a Dunn *post hoc* test with Bonferroni corrections, and Friedman rank sum tests paired with Conover *post hoc* tests with Bonferroni corrections, to test for differences in the average change in steering velocity.

# 3. Results

## (a) Experiment 1: identifying a distorted non-attractive *Odd* chirp

We tested the phonotactic response of females to different chirp envelopes (figure 1*a*, top) to select the least attractive stimulus. A typical recording reveals that this female did not show a pronounced steering response to any of the chirp envelopes; however, she responded strongly to the *Normal* chirp pattern, and walked to the left or right when the corresponding speaker was activated (figure 1*a*). The pooled data show that all chirp envelopes induced some minor steering, as values were not centred on zero, but responses were significantly different overall (Friedman rank sum: $\chi_4^2 = 23.12$, $N = 7$, $p = <0.001$; figure 1*b*). Each type of envelope chirp elicited a significantly lower average steering response when compared to the *Normal* chirp (Rectangle: $z = 4.9$, $p = <0.001$; Oval: $z = 8.16$, $p = <0.001$; Crescendo: $z = 7.35$, $p = <0.001$; Decrescendo: $z = 6.8$, $p = <0.001$). As the Oval envelope chirp had the lowest response ($\bar{x} = 12.45$ mm), it was selected as the *Odd* chirp and used for designing test sequences with different ratios of *Normal* and *Odd* chirps (see Methods). For the *Silent* chirp conditions, the temporal organization of sequences was the same as for the *Odd* chirp condition (figure 1*c*).

## (b) Experiment 2: phonotaxis to combinations of *Normal* chirps and *Odd* or *Silent* chirps

We tested the responses of females to sequences with different ratios of *Normal* chirps combined with *Odd* or *Silent* chirps. The steering response of a female to a sequence of All *Odd* chirps (figure 2*a*, left) shows no obvious steering response, and changes in the animal's walking direction were not coupled to the active speaker. When combinations of *Normal* and *Odd* chirps were presented, the cricket showed no steering behaviour when the ratio between *Normal* and *Odd* chirps was $1:9$, or lower. Steering toward the active speaker started at a ratio of $1:7$, it increased with the fraction of *Normal* chirps increasing, and in this case for the $1:1$ ratio the response was about as strong as the response to All *Normal* chirps (figure 2*a*, right). We found significant differences between steering responses to different ratios (Friedman rank sum: $\chi_8^2 = 111.78$, $N = 27$, $p = <0.001$), with increased directional steering responses when the relative number of *Normal* chirps increased (figure 2*c*, blue bars, table 1). Compared to the sequence with All *Odd* chirps, the sequences with ratios of $1:14$ and all ratios from $1:7$ to $1:1$ elicited a significant stronger phonotactic response (figure 2*c* and table 1). This suggests that a steering response can be elicited when a *Normal* chirp is present in a sequence of *Odd* chirps once every 5.6 s (as in the $1:14$ ratio) or less. Although individuals showed a very strong response to the $1:1$ ratio, the mean response to the sequence with the $1:1$ ratio was 59% of the mean response to an All *Normal* chirp sequence, and significantly lower.

The response of a female to combinations of *Normal* and *Silent* chirps (figure 2*b*) reveals that the insect did not show pronounced steering to any sequence when the ratio between *Normal* and *Silent* chirps was lower than $1:1$. We found a difference in steering responses to different ratios (Friedman rank sum: $\chi_8^2 = 96.713$, $N = 27$, $p = <0.001$), but with less of an increase when compared to the *Odd* chirp condition (figure 2*c*, black and blue bars, table 1). Individuals demonstrated

(a)

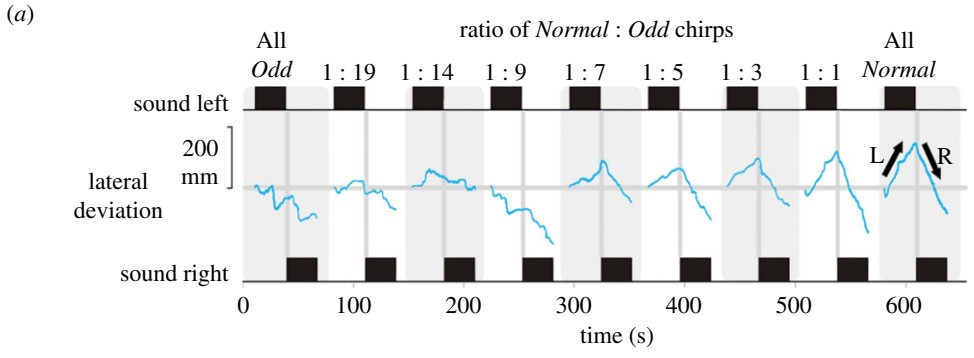

(b)

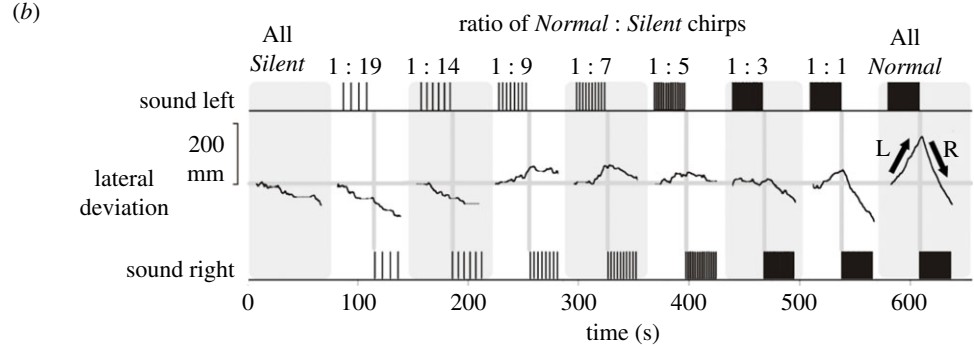

(c) (d)

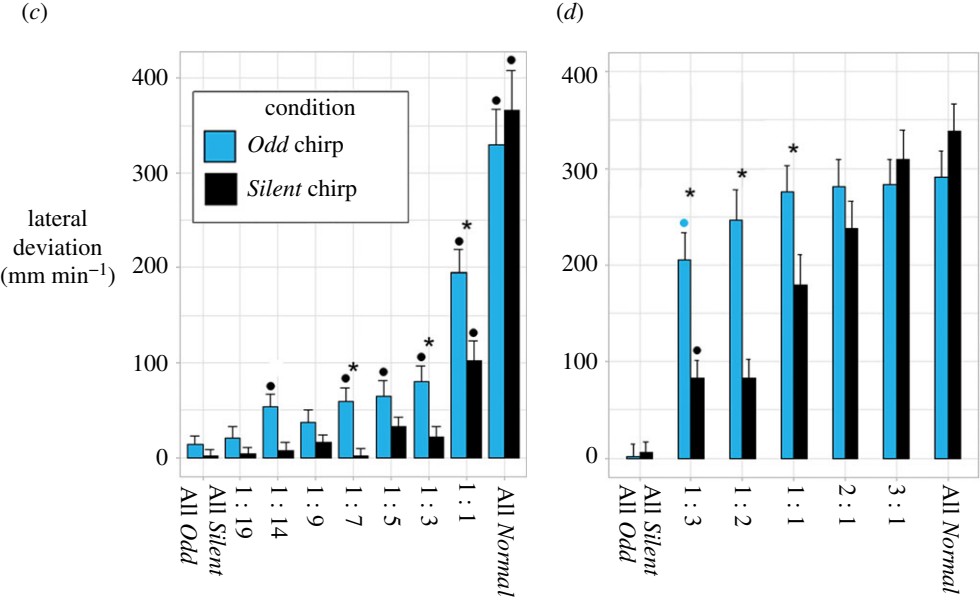

**Figure 2.** Phonotactic steering of a female to different ratios of *Normal* and *Odd* chirps (*a*) and to different ratios of *Normal* and *Silent* chirps (*b*). Arrows indicate steering to the left (L) and right (R). (*c*) Mean lateral deviation toward ratios of *Normal* and *Odd* chirps (blue bars) and *Normal* and *Silent* chirps (black bars) ($\bar{x} \pm$ s.e., $N = 27$). Filled circles indicate a significant difference between a response to that ratio in comparison to the response to a sequence with All *Odd* or All *Silent* chirps. Asterisks indicate a significant difference between the responses of the *Odd* chirp and *Silent* chirp conditions. (*d*) Mean lateral deviation toward the ratios of *Normal* to *Odd* chirps (blue bars) and *Normal* to *Silent* chirps (black bars), where *Normal* chirps occur more frequently ($\bar{x} \pm$ s.e., $N = 27$). Asterisks indicate significant differences between the steering responses in the *Odd* and *Silent* chirp conditions. Filled circles indicate significantly different steering responses to matching tested ratios in (*c*). (Online version in colour.)

significant steering, compared to an All *Silent* chirp sequence, during the 1 : 1 ratio and the All *Normal* chirp sequence (figure 2*c* and table 1). The mean response to the sequence with the 1 : 1 ratio was 28% of the mean response to an All *Normal* chirp sequence, and significantly lower.

In comparison, sequences containing *Odd* chirps elicited significantly stronger steering responses than corresponding *Silent* chirp sequences at the ratios 1 : 14, 1 : 7, 1 : 3 and 1 : 1 (figure 2*c* and table 2). This demonstrates that the *Odd* chirps make a significant contribution to the females steering

behaviour, and that the response to the *Odd* chirps was altered when these were embedded in a sequence of *Normal* chirps. Even sparsely presented *Normal* chirps like in the 1 : 7 or the 1 : 3 sequences, had a substantial impact on phonotactic responses to subsequent *Odd* chirps.

We analysed further ratios of *Normal* to *Odd* or *Silent* chirps to see where full phonotactic responsiveness would occur. Females in this paradigm were tested with ratios of 1 : 3, 1 : 2, 1 : 1, 2 : 1 and 3 : 1 (*Normal* to *Odd or Silent* chirps), along with an All *Odd* or All *Silent* chirp and All

**Table 1.** Differences in lateral deviations between each ratio in the *Odd* and *Silent* acoustic paradigms. *P*-values from *post hoc* Conover tests for pairwise comparisons of directional steering responses between each ratio. Results for both the *Odd* chirp (top-right) and *Silent* chirp (bottom-left) conditions from the first cohort of crickets are shown. *P*-values were calculated with Bonferroni adjusted methods. Italic *p*-values indicate a significant result.

| | All *Odd* or All *Silent* | 1:19 | 1:14 | 1:9 | 1:7 | 1:5 | 1:3 | 1:1 | All *Normal* |
|---|---|---|---|---|---|---|---|---|---|
| All *Odd* or All *Silent* | | 1 | *0.02* | 0.28 | *<0.001* | *<0.001* | *<0.001* | *<0.001* | *<0.001* |
| 1:19 | 1 | | 0.066 | 0.734 | *<0.001* | *<0.001* | *<0.001* | *<0.001* | *<0.001* |
| 1:14 | 1 | 1 | | 1 | 1 | 1 | 0.066 | *<0.001* | *<0.001* |
| 1:9 | 1 | 0.426 | 1 | | 0.28 | 0.28 | *0.004* | *<0.001* | *<0.001* |
| 1:7 | 0.426 | 1 | 0.163 | *0.012* | | 1 | 1 | *<0.001* | *<0.001* |
| 1:5 | 0.163 | *<0.001* | 0.426 | 1 | *<0.001* | | 1 | *<0.001* | *<0.001* |
| 1:3 | 0.057 | *<0.001* | 0.164 | 1 | 1 | 1 | | *<0.001* | *<0.001* |
| 1:1 | *<0.001* | *<0.001* | *<0.001* | *<0.001* | *<0.001* | *<0.001* | *<0.001* | | *0.013* |
| All *Normal* | *<0.001* | *<0.001* | *<0.001* | *<0.001* | *<0.001* | *<0.001* | *<0.001* | *<0.001* | |

*Normal* chirp sequence. In this cohort of crickets, females were more phonotactically active, with increased average responses, in both *Odd* and *Silent* chirp paradigms (figure 2*d*). When comparing corresponding responses between the two cohorts (figure 2*c*), we found that lateral steering in this cohort was significantly higher at the ratio of 1:3, for both the *Odd* and *Silent* chirp sequences (table 3), as in the previously cohort tested.

We found a significant difference in steering among all ratios in both the *Odd* (Friedman rank sum: $X_3^2 = 67.03$, $N = 27$, $p = <0.001$) and *Silent* chirp tests (Friedman rank sum: $X_3^2 = 112.05$, $N = 27$, $p = <0.001$), both showing stronger steering responses as the number of *Normal* chirps increased. Steering responses to sequences with different ratios of *Normal* to *Odd* chirps were not significantly different to each other (figure 2*d*, blue bars, table 4). This suggests that a *Normal* chirp presented once every 1.5 s is sufficient to elicit a maximal steering response when followed by three *Odd* chirps. Steering responses to sequences with different ratios of *Normal* to *Silent* were significantly different between most ratios (figure 2*d*, black bars, table 4). Significant differences occurred between sequences containing *Odd* or *Silent* chirps at the ratios 1:3, 1:2 and 1:1 (figure 2*d* and table 5). Both effects are similar to what we found in the previously tested cohort (figure 2*c*).

## (c) Experiment 3: phonotactic steering velocity in response to *Normal* chirps and *Odd* chirps

The velocity data of the trackball system revealed the fast steering responses toward individual chirps. We averaged the steering velocity for all ratios that showed significantly higher steering than the All *Odd* or All *Silent* chirp sequences (these were 1:14, 1:7, 1:3, 1:1) within Experiment 2. Velocity responses to a given chirp were calculated as the difference between the velocity after the occurrence of the second pulse (or after 60 ms, for *Odd* and *Silent* chirps) and the peak velocity (figure 3*a*).

In the 1:1 ratio, the steering response toward a *Normal* chirp followed by a *Silent* chirp reaches a maximum velocity at around 250 ms and then gradually declines over the next

**Table 2.** Differences in lateral deviations in each ratio between conditions. Output from Wilcoxon signed-rank tests (including sample size, test statistic and *p*-value) on the difference in steering responses to the same ratios between *Odd* chirp and *Silent* chirp conditions form the first cohort of crickets. Italic *p*-values indicate a significant difference.

| ratio | N | Z-statistic | p-value |
|---|---|---|---|
| All *Odd* or All *Silent* | 27 | 237 | 0.258 |
| 1:19 | 27 | 231 | 0.324 |
| 1:14 | 27 | 329 | *<0.001* |
| 1:9 | 27 | 245 | 0.186 |
| 1:7 | 27 | 304 | *0.005* |
| 1:5 | 27 | 259 | 0.095 |
| 1:3 | 27 | 297 | *0.008* |
| 1:1 | 27 | 312 | *0.002* |
| All *Normal* | 27 | 135 | 0.202 |

500 ms to zero as there is no response to the *Silent* chirp (figure 3*a*, black trace). The steering response to the *Normal* chirp is still obvious for the 1:3 ratio, but it becomes smaller with increasing number of *Silent* chirps included in the paradigm. The averaged signals also become noisier, as fewer chirps can be evaluated. Averaging the response to *Normal* chirps followed by *Odd* chirps for the 1:1 ratio reveals again an increased steering velocity in response to the *Normal* chirp but also a clear response to the *Odd* chirp (figure 3*a*, blue trace). It reaches the same peak velocity as the response to the *Normal* chirp but comes with a faster decay. As females steered toward the *Normal* and the *Odd* chirps, the steering velocity signal does not decline to zero. The change in velocity in response to the *Normal* and *Odd* chirp was still obvious for the 1:3 condition and then became weaker with overall decreasing numbers of *Normal* chirps presented.

The change in steering velocity in response to *Normal* chirps in the *Odd* and *Silent* condition reached 4.4 cm s$^{-1}$ and 4.3 cm s$^{-1}$, respectively. The response to the *Odd* chirps reached 3.6 cm s$^{-1}$, and the response to the *Silent* chirps

**Table 3.** Differences in lateral deviations in *Odd* or *Silent* chirp ratios between cohorts. Output includes sample size, test statistic and *p*-value. Italic *p*-values indicate a significant difference.

| | | *Odd* chirp condition comparison | | *Silent* chirp condition comparison | |
|---|---|---|---|---|---|
| ratio | *N* | *Z*-statistic | *p*-value | *Z*-statistic | *p*-value |
| All *Odd* or *Silent* | 54 | 286 | 0.179 | 373 | 0.89 |
| 1 : 3 | 54 | 573 | *<0.001* | 529 | *0.004* |
| 1 : 1 | 54 | 478 | 0.051 | 472 | 0.064 |
| All *Normal* | 54 | 351 | 0.824 | 352 | 0.84 |

**Table 4.** Differences in lateral deviations between additional ratios in the *Odd* and *Silent* acoustic paradigms. *P*-values from *post hoc* Wilcoxon signed-rank pairwise comparisons of directional steering responses between each ratio from the second cohort of crickets. *P*-values were calculated with Bonferroni adjusted methods. Italic *p*-values indicate a significant result.

| | All *Odd* or All *Silent* | 1 : 3 | 1 : 2 | 1 : 1 | 2 : 1 | 3 : 1 | All *Normal* |
|---|---|---|---|---|---|---|---|
| All *Odd* or All *Silent* | | *<0.0001* | *<0.0001* | *<0.0001* | *<0.0001* | *<0.0001* | *<0.0001* |
| 1 : 3 | *0.0056* | | 1 | 0.026 | 0.159 | 0.135 | 0.057 |
| 1 : 2 | *0.0291* | 1 | | 1 | 1 | 1 | 1 |
| 1 : 1 | *<0.0001* | *0.008* | *0.0023* | | 1 | 1 | 1 |
| 2 : 1 | *<0.0001* | *<0.0001* | *<0.0001* | 0.0969 | | 1 | 1 |
| 3 : 1 | *<0.0001* | *<0.0001* | *<0.0001* | *0.0063* | 0.2536 | | 1 |
| All *Normal* | *<0.0001* | *<0.0001* | *<0.0001* | *0.002* | 0.0569 | 1 | |

**Table 5.** Differences in lateral deviations between *Odd* and *Silent* acoustic paradigms when *Normal* chirps are more frequent. Output from appropriate paired tests (including sample size, test statistic and *p*-value). Italic *p*-values indicate a significant difference.

| ratio | *N* | test | statistic | *p*-value |
|---|---|---|---|---|
| All *Odd* or All *Silent* | 27 | paired *t*-test | −0.278 | 0.783 |
| 1 : 3 | 27 | Wilcoxon signed-rank | 579 | *<0.001* |
| 1 : 2 | 27 | Wilcoxon signed-rank | 605 | *<0.001* |
| 1 : 1 | 27 | paired *t*-test | 3.86 | *<0.001* |
| 2 : 1 | 27 | paired *t*-test | 1.973 | 0.059 |
| 3 : 1 | 27 | paired *t*-test | −1.008 | 0.323 |
| All *Normal* | 27 | Wilcoxon signed-rank | 305 | 0.31 |

was only 0.2 cm s$^{-1}$ (figure 3*b*). Changes in average steering velocity differed significantly between the chirp types measured (Kruskal–Wallis: $\chi_4^2 = 67.016$, $N = 54$, $p = >0.001$). Females showed a significant change in steering velocity when presented with *Normal* chirps. They also showed a significant response to *Odd* chirps following a *Normal* chirp, but not to *Silent* chirps following *Normal* chirps (table 6). These data reveal that *Odd* chirps do not just change an overall phonotactic steering bias, when *Odd* chirps are preceded by a *Normal* chirp, they rather elicit a fast steering response similar to the *Normal* chirps.

The previous experiments did not reveal if processing of *Normal* chirps had a bilateral effect. We, therefore, tested females with a 1 : 1 *Normal* to *Odd* ratio where the *Odd* chirps were presented from the opposite side to the *Normal* chirps. Additionally, females were presented with an All *Normal* chirp sequence where each chirp was presented from alternating sides. Individuals showed their ability to rapidly adjust their velocity in response to *Normal* chirps presented from alternating sides (figure 4*a*, black trace) [17]. The same pattern of response was observed to *Odd* chirps as well (figure 4*a*, blue trace), albeit a slightly weaker response to both types of chirps.

We found a significant difference in the change to steering velocity when comparing responses to *Normal* and *Odd* chirps presented from alternating sides, and *Odd* chirps presented in the 30 s control (Friedman rank sum: $X_2^2 = 48.48$, $N = 30$, $p = <0.001$; figure 4*b*). There was no significant difference

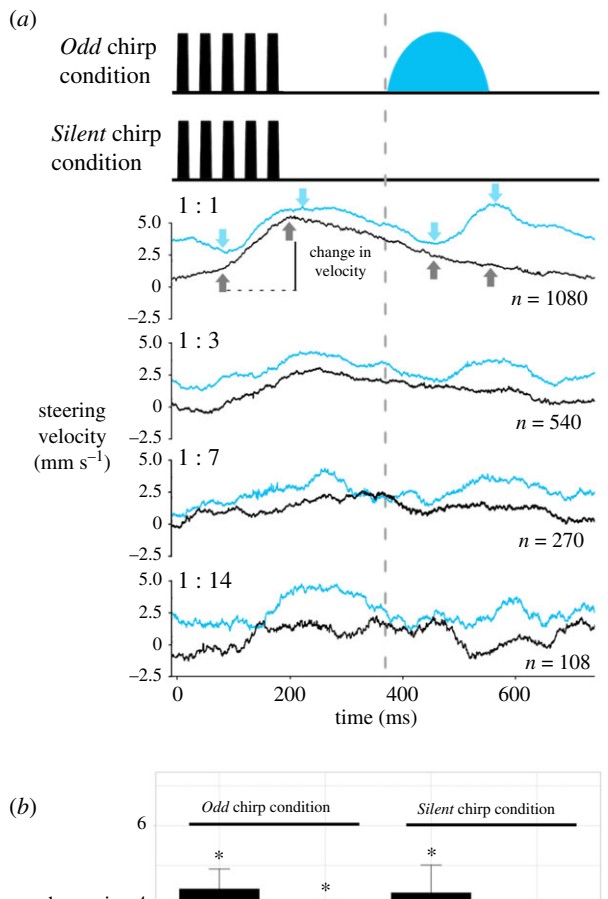

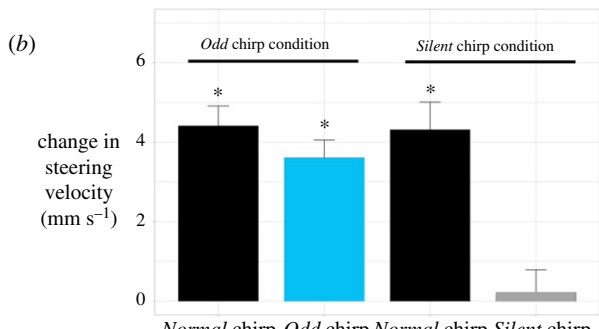

**Figure 3.** (*a*, top) Depiction of the sound pattern used in the *Odd* chirp and *Silent* chirp condition. (*a*, bottom) Averaged steering velocity to a *Normal* chirp followed by either an *Odd* (blue) or a *Silent* chirp, presented for different chirp ratios. Blue lines indicate the response for the *Odd* chirp condition, and black lines show the response to the *Silent* chirp condition. A positive velocity indicates steering toward the active speaker. The chirp period is given by a vertical dotted line. Arrows indicate where measurements of velocity were taken. The start point of the measurements was defined by the end of the second pulse (or after 60 ms, for *Odd* and *Silent* chirps), while the subsequent peak of the response gave the second measurement. For error margins of the averaged data see electronic supplementary material, figure S1. (*b*) Mean change in steering velocity toward *Normal* chirps (black bar), *Odd* chirps (blue bar) and *Silent* chirps (light grey bar), for the 1 : 1 ratio of *Normal* chirps and *Odd* or *Silent* chirps ($\bar{x} \pm$ s.e.). Chirp types with an asterisk indicate responses that are significantly different to the *Silent* chirp in *post hoc* pairwise analyses. (Online version in colour.)

between the response to a *Normal* chirp and a following *Odd* chirp presented from the opposite speaker ($z = 1.23$, $p = 0.327$). Both responses were significantly higher than the change in steering in the All *Odd* control (*Normal* chirp $z = 6.55$, $p = <0.001$; *Odd* chirp $z = -5.32$, $p = <0.001$). These data reveal that the processing of *Normal* chirps presented from one side of the animal also has an impact on the response to *Odd* chirps presented from the opposite side of the animal.

## 4. Discussion

The experiments presented here were based around an odd-ball paradigm used in psychological studies [14], where humans are observed to see how easily they can detect anomalous stimuli. We used this paradigm to test whether female crickets would discriminate against *Odd* chirps or if they would show responses similar to *Normal* chirps. Our results revealed phonotactic steering to distorted non-attractive *Odd* chirps when these were presented in combination with attractive *Normal* chirps. This is evidence that *G. bimaculatus* did not discriminate against the *Odd* chirps but rather employs a pattern recognition system that transiently becomes tolerant to distorted stimuli [23]. Our data reveal that *Odd* chirps, which do not elicit a phonotactic response by their own, will contribute to the phonotactic behaviour when they are interspersed in a sequence of *Normal* chirps. This may allow females during a phonotactic approach in nature to use all chirps of a conspecific calling song for orientation even when the pulse pattern does not meet criteria for pattern recognition [8,9], although the chirp rhythm may still provide a species-specific clue for orientation [24]. Our data are in line with previous studies, demonstrating that non-attractive pulses would transiently elicit phonotactic responses when presented after an ongoing sequence of calling song [25] and that the recognition of the species-specific calling song, at least transiently, alters the female phonotactic responsiveness. Here we show that responses to distorted signals not only occur after a sequence of calling song, but even when the *Odd* chirps are interleaved with the *Normal* chirps, if a sufficient number of *Normal* chirps are still perceived. This is a situation females may likely encounter during phonotactic walking under natural conditions which imposes a more challenging situation on orientation than laboratory-based experiments [26]. It is not clear if the modulation happens at the level of pattern recognition processing or at the motor control of phonotactic steering. As all chirp envelopes tested elicited some weak steering responses, a low-level auditory-to-motor pathway may be present in the cricket nervous system that is upregulated by the pattern recognition process. Steering to *Normal* and *Odd* chirp patterns presented from alternating sides indicates that the processing of *Normal* chirps comes with a bilateral effect on the steering behaviour. This highlights that the response to *Odd* chirps does not represent a general bias of the females to orient toward sound, but that females actively steer toward *Odd* chirps occurring during a sequence of normal song. This had not been observed before [25,27] and may require an additional feature to the well-described concepts of how pattern recognition and directional steering may interact [27].

Besides the implications for pattern recognition tolerant auditory processing has ecological implications. *G. bimaculatus* may have evolved a tolerant auditory system to overcome distortions of acoustic communication signals due to environmental factors. Acoustic signals are prone to degradation due to attenuation and scattering caused by the physical characteristics of the environment [28–30]. Male *G. bimaculatus* signal naturally from burrows [31], and sounds emitted close to the ground are particularly susceptible to distortions [32–34]. The pattern of pulses may become highly distorted [13] and as a result, acoustic signals become degraded before reaching the receiver [10]. By using a tolerant auditory system that is activated by the normal species-specific signal, a receiver may be

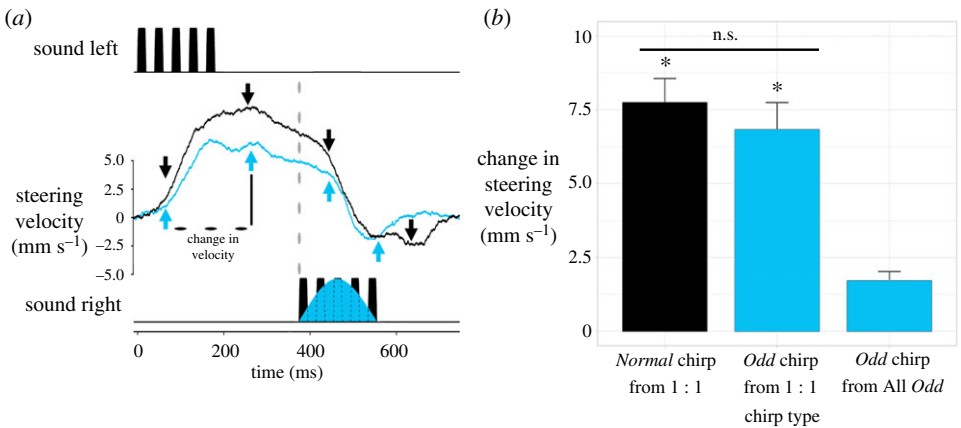

**Figure 4.** (a) Averaged steering velocity to a *Normal* chirp followed by either a *Normal* chirp presented from the opposite side (black trace) or an *Odd* chirp presented from the opposite side (blue trace). Positive velocity indicates steering toward the left speaker and negative velocity indicates steering toward the right speaker. Chirp periods are noted by a vertical dotted line. Arrows indicate where measurements of velocity were taken. The start point of the measurements was defined by the end of the second pulse (or after 60 ms, for *Odd* chirps), while the subsequent peak of the response gave the second measurement. For error margins of the averaged data see electronic supplementary material, figure S2. (b) Mean steering velocity to *Normal* chirps and *Odd* chirps when presented from the opposite side following a *Normal* chirp, and to *Odd* chirps presented just from one side in the All *Odd* test ($\bar{x} \pm$ s.e.). Chirp types with an asterisk indicate responses that are significantly different to the *Odd* chirps from the All *Odd* sequence in *post hoc* pairwise analyses. Bars with a line above indicate responses that were not significantly different to each other. (Online version in colour.)

**Table 6.** Differences of changes in velocity between chirp types. Output from Dunn *post hoc* tests (including test statistic and *p*-value) on pairwise analyses for directional velocity change between chirp types. Data shown are from 1 : 1 ratio. *P*-values calculated with Bonferroni adjusted methods. Italic *p*-values indicate a significant result.

| | *Normal* chirp before *Odd* chirp | *Normal* chirp before *Silent* chirp | *Odd* chirp | *Silent* chirp |
|---|---|---|---|---|
| *Normal* chirp before *Odd* chirp | | $z = -0.444$ | $z = -1.11$ | $z = -5.705$ |
| | | $p = 1$ | $p = 1$ | $p = <0.001$ |
| *Normal* chirp before *Silent* chirp | | | $z = -1.554$ | $z = -5.262$ |
| | | | $p = 0.601$ | $p = <0.001$ |
| *Odd* chirp | | | | $z = -6.816$ |
| | | | | $p = <0.001$ |
| *Silent* chirp | | | | |

able to orient more reliably to a conspecific signal even if it has been distorted by environmental interference. In our study, the presented *Odd* chirps may be considered a distorted and degraded signal, which maintains the same frequency and duration as a chirp but does not feature intra-chirp temporal pulse structures as observed by Simmons [13] for cricket songs measured at a distance to the sender. However, females still responded to these distorted signals as long as a sufficient number of natural chirps were present, highlighting the benefit of a tolerant sensory pathway.

Receivers may also be detecting acoustic signals from other sources simultaneously, such as other signalling individuals [35] or anthropogenic noise [36], which may influence their ability to approach a conspecific signals. In the acridid grasshopper *Chorthippus biguttulus*, broadband background noise presented with the conspecific signals reduces the receivers' ability to respond and orient toward the mate's acoustic signal [37]. In crickets and bushcrickets adaptions of receiver auditory systems prevent such interference by noisy background sound sources from disrupting the neural response to the dominant sender signal. For example, crickets show a gain control mechanism [38,39] that cancels the response to low-level background signals [40].

Our results also have implications for the use of choice experiments to study behavioural responses. Experimental design of mate choice experiments can influence the observed behavioural responses [41]. Two-way choice experiments present animals with signals from two sources and allow the experimental animal to choose between them and to indicate a preference. Such experimental designs can result in stronger mating preferences when compared to no-choice experimental designs [42]. Our data suggest that such a modulation of mating preference could be the result of a tolerant and adaptive sensory system, which might use the information from either stimuli to transiently adjust the attractiveness of the signals, resulting in a biased choice decision. Thus, potential underlying sensory biases need be considered when designing choice experiments [41]. Even no-choice experimental designs could lead to short-term sensory tolerances if stimuli are presented sequentially, without sufficient gaps to allow the modulatory effect of the natural signal to desist. In *Teleogryllus oceanicus*, females may even use remembered acoustic information to shape their phonotactic responses [43].

Due to the COVID-19 pandemic, the crickets used in this study were from different generational cohorts. When we analysed steering responses between cohorts, the second cohort showed overall stronger auditory responses to the

same acoustic stimuli. Nevertheless, the pattern of response remained the same between different cohorts, showing that *Odd* chirps presented after a *Normal* chirp contribute to positive phonotactic responses. This was also apparent in other cohorts used in preliminary tests (unpublished data).

Overall, our data are evidence in support of an auditory system in *G. bimaculatus* that becomes tolerant to distorted signals when processing a conspecific calling song. This is a robust adaptation to the conditions of signal transmission in the field that results in the phonotactic steering to distorted stimuli as long as a *Normal* chirp is presented at least every 5.6 s. Our results have theoretical implications on the neural process of pattern recognition and the resulting control of phonotactic walking behaviour, as pattern recognition shows some form of short-term plasticity.

Ethics. As there are no legal requirements for studies involving orthopteran research subjects in the United Kingdom, no permits or licences were required to house, breed or experiment on *G. bimaculatus*.

Data accessibility. Analyses reported in this article can be reproduced using the data available from the Dryad Digital Repository: https://doi.org/10.5061/dryad.vq83bk3th [44]. The data are provided in electronic supplementary material [45].

Authors' contributions. A.B.: formal analysis, investigation, resources, visualization, writing—original draft, writing—review and editing; B.H.: conceptualization, formal analysis, investigation, methodology, supervision, writing—original draft, writing—review and editing. All authors gave final approval for publication and agreed to be held accountable for the work performed therein.

Competing interests. We have no competing interests.

Funding. Equipment used in this study was funded by the Royal Society and the BBSRC.

Acknowledgements. We thank two final year undergraduate project students, Oliver Bock-Brown and Jack Stockdale, who conducted preliminary experiments that formed the basis of this research. We are grateful to two anonymous reviewers and the manuscripts handling editors for their constructive feedback. Thank you also to members of the Neurobiology of Acoustic Communication Group, who assisted in aspects of experimental design, analysis and interpretation of the results.

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
