## [Peer Review File · Proceedings of the Royal Society B: Biological Sciences]

Review History

RSPB-2021-1889.R0 (Original submission)

Review form: Reviewer 1

Recommendation

Accept with minor revision (please list in comments)

Scientific importance: Is the manuscript an original and important contribution to its field?

Good

General interest: Is the paper of sufficient general interest?

Good

Quality of the paper: Is the overall quality of the paper suitable?

Good

Is the length of the paper justified?

Yes

Should the paper be seen by a specialist statistical reviewer?

Yes

Do you have any concerns about statistical analyses in this paper? If so, please specify them explicitly in your report.

No

It is a condition of publication that authors make their supporting data, code and materials available - either as supplementary material or hosted in an external repository. Please rate, if applicable, the supporting data on the following criteria.

Is it accessible?

Yes

Is it clear?

Yes

Is it adequate?

Yes

Do you have any ethical concerns with this paper?

No

Comments to the Author

See attached file. (See Appendix A)

Review form: Reviewer 2

Recommendation

Accept with minor revision (please list in comments)

Scientific importance: Is the manuscript an original and important contribution to its field?

Good

General interest: Is the paper of sufficient general interest?

Excellent

Quality of the paper: Is the overall quality of the paper suitable?

Excellent

Is the length of the paper justified?

Yes

Should the paper be seen by a specialist statistical reviewer?

No

Do you have any concerns about statistical analyses in this paper? If so, please specify them explicitly in your report.

No

It is a condition of publication that authors make their supporting data, code and materials available - either as supplementary material or hosted in an external repository. Please rate, if applicable, the supporting data on the following criteria.

Is it accessible?

Yes

Is it clear?

Yes

Is it adequate?

Yes

Do you have any ethical concerns with this paper?

No

Comments to the Author

Bent & Hedwig: "Tolerant pattern recognition: Evidence from phonotactic responses in the cricket *Gryllus bimaculatus* (de Geer)", submitted to Proceedings B

In long-range signaling systems such as in acoustic or chemical communication, the broadcast signal of the sender is rarely the one perceived by the receiver. This is due to the effects of the transmission channel, which result in attenuation, frequency filtering and distortions of the amplitude modulation of the original signal. In the current study the authors used the phonotactic responses of female crickets and the ability to measure and quantify phonotaxis with high precision to ask, how tolerant the recognition system in the receiver is to cope with such distortions. The results indicate that crickets have a tolerant pattern recognition system that, once activated by the normal chirp pattern, transiently allows responses to non-attractive, distorted sound patterns.

Minor points:

Line 32: ...move towards males...

Line 41: ...to respond to partially non-attractive...

Line 74:...(Fig. 1A). These envelopes did not contain the pulsed pattern of normal chirps.

Line 240: ...a phonotactic approach in nature...

Discussion:

The problem of signal distortion and its consequence for the receiver pattern recognition has also been studied for the grasshopper *Ch. biguttulus* by B. Ronacher and coworkers. I suggest including some relevant references in the discussion. (e.g. Michelsen and Rohrseitz, 1997; Ronacher and Krahe, 2000; Ronacher and Hoffmann 2003; Reichert and Ronacher 2015)

The reference Hirtenlehner S, Römer H. 2014. Selective phonotaxis of female crickets under natural outdoor conditions. *J Comp Physiol A*. 200:239–250 might be useful here, because they demonstrated that female crickets in a grassland habitat approached the sound source with some larger deviations from a straight path.

Line 274: ...also have implications...

Line 295: ...robust adaptation to the conditions of signal transmission in the field...

Forrest (1994) is cited in References two times (10 and 32), both are incorrect: *AMER. ZOOL.*, 34:644-654 (1994)

Fig. 1B: Although females did not steer to the various types of Odd chirps, they nevertheless walked on the trackball. Why?

Fig. 2C, D: letters for the various ratios are too small.

Fig. 3A: the different chirp ratios could be presented in bold.

Fig. 3B: asterisk missing for the Normal chirp condition?

Fig. 4A: Although correctly explained in the text, the reader could get the impression from the presentation of sound right that Normal and Odd chirps were presented simultaneously and overlapping in time.

Decision letter (RSPB-2021-1889.R0)

06-Oct-2021

Dear Dr Hedwig,

Your manuscript has now been peer reviewed and the reviews have been assessed by an Associate Editor. The reviewers' comments (not including confidential comments to the Editor) and the comments from the Associate Editor are included at the end of this email for your reference. As you will see, the reviewers and the Editors have raised some concerns with your manuscript and we would like to invite you to revise your manuscript to address them.

Research ethics:

Use of animals and field studies:

It is a condition of publication that you make available the data and research materials supporting the results in the article. Please see our Data Sharing Policies (<https://royalsociety.org/journals/authors/author-guidelines/#data>). Datasets should be deposited in an appropriate publicly available repository and details of the associated accession number, link or DOI to the datasets must be included in the Data Accessibility section of the

article (<https://royalsociety.org/journals/ethics-policies/data-sharing-mining/>). Reference(s) to datasets should also be included in the reference list of the article with DOIs (where available).

[http://datadryad.org/submit?journalID=RSPB&manu=\(Document not available\)](http://datadryad.org/submit?journalID=RSPB&manu=(Document%20not%20available)), which will take you to your unique entry in the Dryad repository.

Please submit a copy of your revised paper within three weeks. If we do not hear from you within this time your manuscript will be rejected. If you are unable to meet this deadline please let us know as soon as possible, as we may be able to grant a short extension.

Best wishes,
Professor Loeske Kruuk
mailto: proceedingsb@royalsociety.org

Associate Editor

Board Member: 1

Comments to Author:

Thank you for your submission. Two referees have kindly reviewed the manuscript. Both are supportive, providing constructive suggestions to clarify some aspects of the methods and results. The authors should consider referee 1's query on whether the figures support interpretation in places. Referee 2 offers useful suggestions to set the work more fully in the context of the literature.

Extending from referee 2's point, for the broad audience of the journal I find the introduction and discussion clearly written but quite specialist to the study system. I would suggest setting the scene more generally around line 36 by expanding a little on the introduction to acoustic signal degradation (e.g. moving up the point at line 270-273), and perhaps by adding a brief introduction to tolerance in pattern recognition systems per se (e.g. if there are examples where this has been observed in other sensory systems/taxa). I suggest some minor changes to the

abstract: first, adding a sentence to explain that stimulus design mimics signal distortion with distance. Second (potentially throughout), using a more specific term than 'non-attractive, odd' (as this can be read as related to male quality rather than signal degradation, and the oddball paradigm isn't yet introduced). Third, 'silent chirp' is unclear - would 'silent control' be more appropriate? Last, the opening and concluding sentences could be more specific - referee 1's phrasing may be useful ('as long as a sufficient number of natural signals are present'). In the discussion lines 244, 253 and paragraph 273, it would be useful to contrast more directly to reference 25 to show how the work advances understanding. Please also remember the general audience when introducing specialist methods, e.g. line 46 'air suspended trackball set-up'.

Reviewer(s)' Comments to Author:

Referee: 1

Comments to the Author(s)

See attached file.

Referee: 2

Comments to the Author(s)

Bent & Hedwig: "Tolerant pattern recognition: Evidence from phonotactic

responses in the cricket *Gryllus bimaculatus* (de Geer)", submitted to Proceedings B

In long-range signaling systems such as in acoustic or chemical communication, the broadcast signal of the sender is rarely the one perceived by the receiver. This is due to the effects of the

transmission channel, which result in attenuation, frequency filtering and distortions of the amplitude modulation of the original signal. In the current study the authors used the

phonotactic responses of female crickets and the ability to measure and quantify phonotaxis with high precision to ask, how tolerant the recognition system in the receiver is to cope with such

distortions. The results indicate that crickets have a tolerant pattern recognition system that, once activated by the normal chirp pattern, transiently allows responses to non-attractive, distorted sound patterns.

Minor points:

Line 32: ...move towards males...

Line 41: ...to respond to partially non-attractive...

Line 74:...(Fig. 1A). These envelopes did not contain the pulsed pattern of normal chirps.

Line 240: ...a phonotactic approach in nature...

Discussion:

The problem of signal distortion and its consequence for the receiver pattern recognition has also been studied for the grasshopper *Ch. biguttulus* by B. Ronacher and coworkers. I suggest

including some relevant references in the discussion. (e.g. Michelsen and Rohrseitz, 1997;

Ronacher and Krahe, 2000; Ronacher and Hoffmann 2003; Reichert and Ronacher 2015)

The reference Hirtenlehner S, Römer H. 2014. Selective phonotaxis of female crickets under natural outdoor conditions. *J Comp Physiol A*. 200:239–250 might be useful here, because they

demonstrated that female crickets in a grassland habitat approached the sound source with some larger deviations from a straight path.

Line 274: ...also have implications...

Line 295: ...robust adaptation to the conditions of signal transmission in the field...

Forrest (1994) is cited in References two times (10 and 32), both are incorrect: *AMER. ZOOL.*, 34:644-654 (1994)

Fig. 1B: Although females did not steer to the various types of Odd chirps, they nevertheless walked on the trackball. Why?

Fig. 2C, D: letters for the various ratios are too small.

Fig. 3A: the different chirp ratios could be presented in bold.

Fig. 3B: asterisk missing for the Normal chirp condition?

Fig. 4A: Although correctly explained in the text, the reader could get the impression from the presentation of sound right that Normal and Odd chirps were presented simultaneously and overlapping in time.

Author's Response to Decision Letter for (RSPB-2021-1889.R0)

See Appendices B & C.

Decision letter (RSPB-2021-1889.R1)

18-Nov-2021

Dear Dr Hedwig

I am pleased to inform you that your Review manuscript RSPB-2021-1889.R1 entitled "Tolerant pattern recognition: Evidence from phonotactic responses in the cricket &em>Gryllus bimaculatus&/em> (de Geer)" has been accepted for publication in Proceedings B.

The referee(s) do not recommend any further changes. Therefore, please proof-read your manuscript carefully and upload your final files for publication. Because the schedule for publication is very tight, it is a condition of publication that you submit the revised version of your manuscript within 7 days. If you do not think you will be able to meet this date please let me know immediately.

In the Data Accessibility section - please add the Dryad DOI for your data.

To upload your manuscript, log into <http://mc.manuscriptcentral.com/prsb> and enter your Author Centre, where you will find your manuscript title listed under "Manuscripts with Decisions." Under "Actions," click on "Create a Revision." Your manuscript number has been appended to denote a revision.

You will be unable to make your revisions on the originally submitted version of the manuscript. Instead, upload a new version through your Author Centre.

- 1) A text file of the manuscript (doc, txt, rtf or tex), including the references, tables (including captions) and figure captions. Please remove any tracked changes from the text before submission. PDF files are not an accepted format for the "Main Document".
- 2) A separate electronic file of each figure (tiff, EPS or print-quality PDF preferred). The format should be produced directly from original creation package, or original software format. Please note that PowerPoint files are not accepted.
- 3) Electronic supplementary material: this should be contained in a separate file from the main text and the file name should contain the author's name and journal name, e.g `authorname_procb_ESM_figures.pdf`

All supplementary materials accompanying an accepted article will be treated as in their final form. They will be published alongside the paper on the journal website and posted on the online figshare repository. Files on figshare will be made available approximately one week before the

accompanying article so that the supplementary material can be attributed a unique DOI. Please see: <https://royalsociety.org/journals/authors/author-guidelines/>

4) Data-Sharing and data citation

It is a condition of publication that data supporting your paper are made available. Data should be made available either in the electronic supplementary material or through an appropriate repository. Details of how to access data should be included in your paper. Please see <https://royalsociety.org/journals/ethics-policies/data-sharing-mining/> for more details.

If you wish to submit your data to Dryad (<http://datadryad.org/>) and have not already done so you can submit your data via this link <http://datadryad.org/submit?journalID=RSPB&manu=RSPB-2021-1889.R1> which will take you to your unique entry in the Dryad repository.

Once again, thank you for submitting your manuscript to Proceedings B and I look forward to receiving your final version. If you have any questions at all, please do not hesitate to get in touch.

Sincerely,
Professor Loeske Kruuk
Editor, Proceedings B
<mailto:proceedingsb@royalsociety.org>

Associate Editor Board Member
Comments to Author:
(There are no comments.)

Reviewer(s)' Comments to Author:

Decision letter (RSPB-2021-1889.R2)

22-Nov-2021

Dear Dr Hedwig

I am pleased to inform you that your manuscript entitled "Tolerant pattern recognition: Evidence from phonotactic responses in the cricket *Gryllus bimaculatus* (de Geer)" has been accepted for publication in Proceedings B.

Data Accessibility section

Open Access

Paper charges

Sincerely,

Proceedings B

Appendix A

Summary

Bent & Hedwig report that female *Gryllus bimaculatus* crickets perform phonotaxis toward an otherwise unattractive acoustic stimulus when it is interleaved with an attractive signal from the calling song of conspecific males. They vary the ratio of unattractive (“odd”) and conspecific (“normal”) signals and find that females will steer toward a ratio of at least 1 normal to 7 odd signals. Since the steering responses to these interleaved stimuli are greater than the corresponding stimuli with a silent period instead of odd signals, the authors conclude that the presence of the odd stimulus must influence female behavior. This pattern could result from either an increase in overall responsiveness to the normal signals or from active steering toward the odd signals. To test whether females are indeed navigating toward the odd signal, the authors measure steering responses to normal and the following odd signal, and find no difference in response magnitude. Thus, despite not responding to the odd signals when presented alone, females will walk toward the unattractive odd signal when it is presented after a normal signal. The authors show that this effect occurs bilaterally, with a normal signal presented to one side increasing the steering response to an odd signal presented to the opposite side. Together, these results suggest tolerance in the pattern recognition system which allows for some deviation from the conspecific pattern, as long as a sufficient number of natural signals are present. The results have implications for understanding acoustic pattern recognition across auditory systems.

Major Concerns

1. From Fig. 1A and the Methods, the structure of the odd chirps relative to the normal chirp is not clear. Are the amplitude modulations applied to the normal chirp structure (if so, how is a rectangular envelope different from normal?), or does each odd chirp contain essentially one long pulse with the described amplitude modulations? It is not until the Discussion (line 266) that the authors clarify this point. It would be helpful to mention sooner.
2. In the Methods, the authors should provide more details about how the steering responses were analyzed. For instance, how was the one-minute window analyzed in Fig. 1B defined? How were the points for velocity measurements determined in Fig. 3A and 4A?
3. The authors use the findings in Fig. 2D to interpret that a normal chirp every 1.5 sec is sufficient to elicit a maximal steering response (line 182). This seems to contradict the findings in Fig. 2C, where responses to 1:3 odd chirps are less than 1/3 the responses to all normal chirps.

Other Concerns

1. In the Methods, the authors should provide details of their sound delivery system.
2. In the Results, the authors contradict themselves when they first write that in Fig. 2A, the exemplar steering response to 1:1 ratio was about as strong as the response to all Normal chirps (Lines 146-147), but then the population average was significantly lower (lines 154-155). This was confusing.
3. In Experiment 1, the authors’ rationale for concluding that the response to the Odd chirps was altered when embedded in a sequence of Normal chirps (line 167) is not

clear. Could it alternatively be the case that the steering response to the (infrequent) Normal chirps is enhanced by the presence of the Odd chirps?

4. The authors find steering responses to the first odd chirp following a normal chirp in Fig. 3A. Would analyzing responses to subsequent odd chirps (as the different ratios allow) provide an estimate of the window of steering response enhancement following a normal chirp?
5. In Fig. 3A, the vertical dotted line is difficult to see.
6. In Fig. 3B, the authors are missing an asterisk over the normal chirp in the silent chirp condition (according to Table 6).
7. The authors should include standard deviation for the averaged traces in Fig. 3A and 4A.

Appendix B

Dr habil Berthold Hedwig
Reader in Neurobiology

Cambridge 9. Nov 2021

To the Editors of Proceedings of the Royal Society B
Dear Prof Kruuk

Thank you for reviewing our MS entitled "*Tolerant pattern recognition: Evidence from phonotactic responses in the cricket *Gryllus bimaculatus* (de Geer)*". We are very grateful for the constructive comments and advice that you and the referees provided, which improved the quality of the manuscript.

We have dealt and addressed all issues of concern as described in our detailed response and we hope that this will allow you to consider our work for publication.

Yours sincerely,

Adam Bent and Berthold Hedwig

Appendix C

Associate Editor / Board Member: 1

Comments to Author:

Thank you for your submission. Two referees have kindly reviewed the manuscript. Both are supportive, providing constructive suggestions to clarify some aspects of the methods and results. The authors should consider referee 1's query on whether the figures support interpretation in places. Referee 2 offers useful suggestions to set the work more fully in the context of the literature.

Extending from referee 2's point, for the broad audience of the journal I find the introduction and discussion clearly written but quite specialist to the study system. I would suggest setting the scene more generally around line 36 by expanding a little on the introduction to acoustic signal degradation (e.g. moving up the point at line 270-273),

We have added at line 36: Due to reverberations, the calling song's pulse pattern may become degraded within a short distance from the signaller, and conspecifics no longer respond to such distorted signals (Simmons, 1988), as they become difficult to process by a pattern recognition system.

and perhaps by adding a brief introduction to tolerance in pattern recognition systems per se (e.g. if there are examples where this has been observed in other sensory systems/taxa).

We think that the term "tolerance" describes the situation very well. To the best of our knowledge we are not aware of any other examples in the literature, which use the same terminology.

I suggest some minor changes to the abstract: first, adding a sentence to explain that stimulus design mimics signal distortion with distance.

We added: ...analysed tolerance in auditory steering responses to "Odd" chirps, mimicking a signal distorted by the transmission channel, and control "Silent" chirps by ...

We also added to the Methods, Experiment 1: Crickets do not respond to conspecific calling songs with the pulse structure of chirps distorted by reverberations (Simmons 1988). In order to create a distorted non-attractive chirp pattern we designed artificial chirp stimuli with different amplitude envelopes that maintained the same chirp period, duration and frequency as the Normal chirps, but had no pulse structure.

Second (potentially throughout), using a more specific term than 'non-attractive, odd' (as this can be read as related to male quality rather than signal degradation, and the oddball paradigm isn't yet introduced).

According to your advice, we twice removed the term "non-attractive" from the abstract, and now introduce it as a "distorted non-attractive" sound signal. We also changed the text to either "distorted non-attractive" or just "distorted".

Third, 'silent chirp' is unclear – would 'silent control' be more appropriate?

In this case we rather would like to keep our terminology, as this refers to the identical time scale of the signal presentation. We now say in the text, that we test the response to ...control "Silent" chirps by ...

Last, the opening and concluding sentences could be more specific - referee 1's phrasing may be useful ('as long as a sufficient number of natural signals are present').

We had already indicated in the text that pattern recognition needs to be activated, however to make the point more clear we followed your suggestion and rephrased the sentence to:once activated, transiently allows responses to distorted non-attractive sound patterns, as long as a sufficient number of natural chirps are present.

We also now highlight this point in the Discussion: However, females still responded to these distorted signals as long as a sufficient number of natural chirps were present, highlighting the benefit of a tolerant sensory pathway.

Formatted: Indent: First line: 0 cm, Line spacing: single

In the discussion lines 244, 253 and paragraph 273, it would be useful to contrast more directly to reference 25 to show how the work advances understanding.

Deleted: ¶

To differentiate more clearly, we added: Here we show that responses to distorted signals not only occur after a substantial sequence of calling song, but even when the Odd chirps are interleaved with the Normal chirps, if a sufficient number of Normal chirps are still perceived. This is a situation female may likely encounter during phonotactic walking under natural conditions, which imposes a more challenging situation on orientation than lab-based experiments (Hirtenlehner and Roemer 2014).

Deleted: .

Deleted: .

Formatted: Not Highlight

And later we say: but that females actively steer towards Odd chirps occurring during a sequence of normal song.

We also point out that the experiments by Ronacher are based on an increased background noise level, and that the auditory system of crickets may reduce the impact of background noise.

Please also remember the general audience when introducing specialist methods, e.g. line 46 'air suspended trackball set-up'.

Deleted: ¶
¶

We rephrased this part of the Introduction to: We tethered crickets so that they walked on an air suspended trackball, while sound patterns were presented from a speaker at the left or right side [Hedwig and Poulet 2005]. The rotations of the trackball were measured and revealed the female phonotactic steering behaviour to sound sequences with attractive and distorted non-attractive chirps combined at different ratios.

Formatted: Indent: First line: 0 cm, Space After: 0 pt, Line spacing: single, Don't adjust space between Latin and Asian text, Don't adjust space between Asian text and numbers

Formatted: Font color: Custom Color(50,49,48), Pattern: Clear (White)

Deleted: .

Deleted: Also

Also, in the Methods we now give more detail regarding the trackball system: Females tethered to an insect pin, were placed stationary on top of the lightweight air suspended track ball, which they moved with their legs when walking. The forward-backward and left-right movements of the trackball were captured with an optoelectronic sensor and indicated the walking speed and direction of the specimen.

Deleted: .

Reviewer(s)' Comments to Author:

Formatted: Font: Bold

Referee: 1

Formatted: Font: Bold

Summary

Bent & Hedwig report that female *Gryllus bimaculatus* crickets perform phonotaxis toward an otherwise unattractive acoustic stimulus when it is interleaved with an attractive signal from the calling song of conspecific males. They vary the ratio of unattractive ("odd") and conspecific ("normal") signals and find that females will steer toward a ratio of at least 1 normal to 7 odd signals. Since the steering responses to these interleaved stimuli are greater than the corresponding stimuli with a silent period instead of odd signals, the authors conclude that the presence of the odd stimulus must influence female behavior. This pattern could result from either an increase in overall responsiveness to the normal signals or from active steering toward the odd signals. To test whether females are indeed navigating toward the odd signal, the authors measure steering responses to normal and the following odd signal, and find no difference in response magnitude. Thus, despite not responding to the odd signals when presented alone, females will walk toward the unattractive

odd signal when it is presented after a normal signal. The authors show that this effect occurs bilaterally, with a normal signal presented to one side increasing the steering response to an odd signal presented to the opposite side. Together, these results suggest tolerance in the pattern recognition system which allows for some deviation from the conspecific pattern, as long as a sufficient number of natural signals are present. The results have implications for understanding acoustic pattern recognition across auditory systems.

Major Concerns

1. From Fig. 1A and the Methods, the structure of the odd chirps relative to the normal chirp is not clear. Are the amplitude modulations applied to the normal chirp structure (if so, how is a rectangular envelope different from normal?), or does each odd chirp contain essentially one long pulse with the described amplitude modulations? It is not until the Discussion (line 266) that the authors clarify this point. It would be helpful to mention sooner.

When describing Experiment 1 we now make it clear that *the artificial chirps had no pulse structure and that the envelopes covered one chirp-long sound.*

Formatted: Font: Italic

Formatted: Font: Italic

2. In the Methods, the authors should provide more details about how the steering responses were analyzed. For instance, how was the one-minute window analyzed in Fig. 1B defined?

In the Data analysis we added: Each chirp type was presented for 30s from the left and right. The steering velocity of the crickets was calculated based on the output of the optical sensor and the lateral deviation over each presentation was obtained by integrating the velocity data for the left and right sound presentation. These were pooled to obtain a measure for the phonotactic response over a one-minute window [Hedwig and Poulet 2005].

Formatted: Line spacing: single

How were the points for velocity measurements determined in Fig. 3A and 4A?

We added to the legends of Fig. 3A and Fig 4: The start point of the measurements was defined by the latency of the steering responses, while the subsequent peak of the response gave the second measurement.

3. The authors use the findings in Fig. 2D to interpret that a normal chirp every 1.5 sec is sufficient to elicit a maximal steering response (line 182). This seems to contradict the findings in Fig. 2C, where responses to 1:3 odd chirps are less than 1/3 the responses to all normal chirps.

The referee points to a discrepancy that we also had noticed, as we were using two different cohorts of crickets due to the Covid situation. Besides the same husbandry conditions, the second cohort of crickets demonstrated a much higher responsiveness than the first cohort. We have no explanation for this. In the Results we changed the text to: When comparing corresponding responses between the two cohorts (Figure 2C), we found that lateral steering in this cohort was significantly higher at the ratio of 1 : 3, for both the Odd and Silent chirp sequences (Table 5), as in the previously cohort tested.

Deleted: ¶
¶

Formatted: Font: Not Bold

Formatted: Font: Not Bold

Other Concerns

1. In the Methods, the authors should provide details of their sound delivery system.

In the Methods we added: All acoustic stimuli were delivered via two speakers (Neo 13S, Sinus Live, Conrad Electronics, Hirschau, Germany) positioned 57 cm in front of the cricket at the left and right side at 45 deg to its long axis.

Formatted: Line spacing: single

2. In the Results, the authors contradict themselves when they first write that in Fig. 2A, the exemplar steering response to 1:1 ratio was about as strong as the response to all Normal chirps

(Lines 146-147), but then the population average was significantly lower (lines 154-155). This was confusing.

This is due to variability of the behavioural data. Some females responded to the 1:1 ratio very strongly, but in the mean the response to the 1:1 ratio was 59% of the response to an All Normal sequence.

We altered the text accordingly to point this out.

3. In Experiment 1, the authors' rationale for concluding that the response to the Odd chirps was altered when embedded in a sequence of Normal chirps (line 167) is not clear. Could it alternatively be the case that the steering response to the (infrequent) Normal chirps is enhanced by the presence of the Odd chirps?

This consideration is addressed by averaging the steering velocity in Fig. 3, demonstrating the additional steering response to the *Odd* chirps.

Formatted: Font: Italic

4. The authors find steering responses to the first odd chirp following a normal chirp in Fig. 3A. Would analyzing responses to subsequent odd chirps (as the different ratios allow) provide an estimate of the window of steering response enhancement following a normal chirp?

Before selecting the time window for averaging, we looked into this option, however the data indicating the responses to the 2nd or 3rd *Odd* chirp after a normal chirp become very noisy with the sound patterns presented. We only see a clear effect immediately after the normal chirp, which decreases with the ratio of *Normal* to *Odd* chirps becoming smaller. Insight into the dynamics of the response modulation had been provided by Poulet and Hedwig (2005) when non-attractive sound pulses were presented after female crickets had been exposed to 10 s sequences of calling song.

Deleted: ¶

Formatted: Superscript

Formatted: Superscript

Formatted: Font: Italic

Formatted: Font: Italic

Formatted: Font: Italic

5. In Fig. 3A, the vertical dotted line is difficult to see. - Edited

Deleted: ¶

6. In Fig. 3B, the authors are missing an asterisk over the normal chirp in the silent chirp condition (according to Table 6). - Corrected

Formatted: Font color: Blue

Formatted: Font color: Blue

7. The authors should include standard deviation for the averaged traces in Fig. 3A and 4A

Including the standard deviation would distort the lay-out of these figures, we therefore give more detailed information of the averages with the SEM in the supplementary material. We included a note in the corresponding figure legends.

Referee: 2

Formatted: Font: Bold

Comments to the Author(s)

Bent & Hedwig: "Tolerant pattern recognition: Evidence from phonotactic responses in the cricket *Gryllus bimaculatus* (de Geer)", submitted to Proceedings B

In long-range signaling systems such as in acoustic or chemical communication, the broadcast signal of the sender is rarely the one perceived by the receiver. This is due to the effects of the transmission channel, which result in attenuation, frequency filtering and distortions of the amplitude modulation of the original signal. In the current study the authors used the phonotactic responses of female crickets and the ability to measure and quantify phonotaxis with high precision to ask, how tolerant the recognition system in the receiver is to cope with such distortions. The

results indicate that crickets have a tolerant pattern recognition system that, once activated by the normal chirp pattern, transiently allows responses to non-attractive, distorted sound patterns.

Minor points:

Line 32: ...move towards males... corrected

Formatted: Font color: Blue

Line 41: ...to respond to partially non-attractive...

We now say: We investigated the propensity of crickets to respond to distorted non-attractive acoustic

Formatted: Space After: 0 pt, Line spacing: single

Line 74:...(Fig. 1A). These envelopes did not contain the pulsed pattern of normal chirps. Inserted into the text

Line 240: ...a phonotactic approach in nature... - Added

Formatted: Font color: Blue

Discussion:

The problem of signal distortion and its consequence for the receiver pattern recognition has also been studied for the grasshopper *Ch. biguttulus* by B. Ronacher and coworkers. I suggest including some relevant references in the discussion. (e.g. Michelsen and Rohrseitz, 1997; Ronacher and Krahe, 2000; Ronacher and Hoffmann 2003; Reichert and Ronacher 2015)

The paper by Ronacher related to the effect of masking background noise on recognition and orientation have been added

Formatted: Space After: 0 pt

The reference Hirtenlehner S, Römer H. 2014. Selective phonotaxis of female crickets under natural outdoor conditions. *J Comp Physiol A*. 200:239–250 might be useful here, because they demonstrated that female crickets in a grassland habitat approached the sound source with some larger deviations from a straight path.

The reference to Hirtenlehner and Roemer is given in the Discussion.

Line 274: ...also have implications... - Added

Formatted: Font color: Accent 1

Line 295: ...robust adaptation to the conditions of signal transmission in the field... - Added

Formatted: Font color: Accent 1

Forrest (1994) is cited in References two times (10 and 32), both are incorrect: AMER. ZOOL., 34:644-654 (1994) - Corrected

Formatted: Font color: Accent 1

Fig. 1B: Although females did not steer to the various types of Odd chirps, they nevertheless walked on the trackball. Why?

This is beyond our control and related the animals "motivation". While some females do not start walking, spontaneous walking on the trackball without acoustic stimulation is normal, in the presence of attractive sound stimuli females then may orient towards the acoustic stimulus.

Formatted: Space After: 0 pt

Deleted: .

Fig. 2C, D: letters for the various ratios are too small. This has been altered

Fig. 3A: the different chirp ratios could be presented in bold. - Edited

Fig. 3B: asterisk missing for the Normal chirp condition? - Corrected

Fig. 4A: Although correctly explained in the text, the reader could get the impression from the presentation of sound right that Normal and Odd chirps were presented simultaneously and overlapping in time.

We included and *either-or* in the Fig legend to make clearer: Averaged steering velocity to a *Normal* chirp followed by either a *Normal* chirp presented from the opposite side (black trace) or an *Odd* chirp presented from the opposite side (blue trace).

Formatted: Font color: Accent 1

Formatted: Font color: Accent 1

Formatted: Space After: 0 pt

Formatted: Font: Italic